# On $K^*$ Search for Top-$k$ Planning

**Junkyu Lee, Michael Katz, Shirin Sohrabi**

IBM T.J. Watson Research Center
1101 Kitchawan Rd, Yorktown Heights, NY 10598, USA
{junkyu.lee, michael.katz1}@ibm.com, ssohrab@us.ibm.com

## Abstract

Finding multiple high-quality plans is essential in many planning applications, and top-$k$ planning asks for finding the $k$ best plans, naturally extending cost-optimal classical planning. Several attempts have been made to formulate top-$k$ classical planning as a $k$-shortest paths finding problem and apply $K^*$ search, which alternates between $A^*$ and Eppstein's algorithm. However, earlier work had shortcomings, among which were failing to handle inconsistent heuristics and degraded performance in Eppstein's algorithm implementations. As a result, existing evaluation results severely underrate the performance of the $K^*$ based approach to top-$k$ planning. In this paper, we present a new top-$k$ planner based on a novel variant of $K^*$ search. We address the following three aspects. First, we show an alternative implementation of Eppstein's algorithm for classical planning, which resolves a major bottleneck in earlier attempts. Second, we present a new strategy for alternating $A^*$ and Eppstein's algorithm, that improves the performance of $K^*$ on the classical planning benchmarks. Last, we introduce a simple mitigation of the limitation of $K^*$ to tasks with a single goal state, allowing us to preserve heuristic informativeness in face of imposed task reformulation. Empirical evaluation results show that the proposed approach achieves the state-of-the-art performance on the classical planning benchmarks. The code is available at https://github.com/IBM/kstar.

## Introduction

Enumerating the $k$-best solutions is desirable for designing many optimization algorithms and it is essential for various real-world applications (Eppstein 2014). Top-$k$ planning (Riabov, Sohrabi, and Udrea 2014; Katz et al. 2018) enumerates the $k$ best plans in cost optimal classical planning problems, and top-$k$ planners have been deployed in many planning applications such as hypothesis generation (Sohrabi, Riabov, and Udrea 2016), scenario planning (Sohrabi et al. 2018), and machine learning pipeline generation (Katz et al. 2020), to name a few. To date, top-$k$ planners implement one of the following three approaches: (1) adapting $K^*$ (Aljazzar and Leue 2011) for classical planning (Riabov, Sohrabi, and Udrea 2014; Katz et al. 2018), (2) iteratively reformulating the input task to forbid found plans (FI) (Riabov, Sohrabi, and Udrea 2014; Katz et al. 2018), or (3) extending symbolic

search (SymK) (Speck, Mattmüller, and Nebel 2020). In this paper, we focus on top-$k$ planning with $K^*$ search which finds $k$ shortest *non-simple* paths from a source node to a terminal node in a search graph by alternating between $A^*$ (Hart, Nilsson, and Raphael 1968) for exploring the search graph $G_{A^*}$ and Eppstein's $k$ shortest paths algorithm ($EA$) (Eppstein 1998) for enumerating $k$ shortest non-simple paths in $G_{A^*}$. The merit of $EA$ is in its asymptotic time complexity: $O(n \log n + m + k)$, where $n$ is the number of nodes and $m$ is the number of edges in a search graph, namely constant time overhead per path compared to finding a single shortest path. However, such a remarkable performance is achievable only after using complicated heap-based data structures, often limiting broader usage. This makes simpler alternatives, such as $m$-best search (Dechter, Flerova, and Marinescu 2012) appealing in complex search spaces. In practice, Jiménez and Marzal (1999) reported that $EA$ may spend a significant amount of time for initializing necessary data structures, and Jiménez and Marzal (2003) improved the running time by building them in a lazy manner. The intensive memory requirement for storing all explored edges, as well as all nodes, is another shortcoming which $K^*$ (Aljazzar and Leue 2011) alleviates by expanding search graphs on the fly using $A^*$.

In classical top-$k$ planning, Riabov, Sohrabi, and Udrea (2014) implemented a blind $K^*$ search in the SPPL planner (Riabov and Liu 2005). Later, Katz et al. (2018) implemented another variant of $K^*$ based top-$k$ planner that supports consistent heuristics in the Fast Downward planning system (Helmert 2006a). When we examine earlier evaluation results, we see that existing $K^*$ based top-$k$ planners could not utilize inconsistent heuristics such as LM-cut (Helmert and Domshlak 2009), which are known to be among the most informative and best-performing ones. In addition, Katz et al. (2018) reported non-anytime behavior of $K^*$ based planners, namely, finding all request $k$ plans within a short time or failing to find any. Further, Speck, Mattmüller, and Nebel (2020) reported that the performance improvement of $K^*$ based planners due to iPDB heuristics (Haslum et al. 2007) compared with the blind heuristics was less than two percents in coverage. Although evaluation results may vary depending on the evaluation settings and benchmark sets used, the existing results clearly motivate revisiting $K^*$ search for top-$k$ planning. In this paper,

we address the following three aspects:

- an alternative implementation of lazy $EA$ using linked lists to utilize pointer-based operations,
- a new switching strategy between $A^*$ and $EA$ that reflects the in-depth evaluation of various combinations of hyperparameters that trigger switching between the two,
- a simple method that preserves heuristics informativeness despite the need to reformulate the input classical planning task as a single-goal state task.

We implemented a new $K^*$ based top-$k$ planner in the `Fast Downward` planning system release 20.6 (Helmert 2006a), and evaluated on all benchmarks from optimal tracks of International Planning Competitions 1998–2018 along with state-of-the-art top-$k$ planners, FI (Katz et al. 2018) and SymK (Speck, Geiser, and Mattmüller 2020). The evaluation results show that the $K^*$ based approach was indeed underrated in the earlier evaluations, and we see that the $K^*$ based approach performs better than FI and SymK.

## Background

This section introduces top-$k$ planning and necessary notations and basic concepts in $K^*$ search for top-$k$ planning.

### Top-$k$ Classical Planning

A *planning task* $\Pi = \langle \mathcal{V}, \mathcal{O}, s_0, s_\star \rangle$ in SAS$^+$ formalism (Bäckström and Nebel 1995) consists of a finite set of finite-domain *state variables* $\mathcal{V}$, a finite set of *actions* $\mathcal{O}$, an *initial state* $s_0$, and the *goal* $s_\star$. Each variable $v \in \mathcal{V}$ is associated with a finite domain $dom(v)$ of values. An assignment of a value $d \in dom(v)$ to a variable $v \in \mathcal{V}$ denoted by a pair $\langle v, d \rangle$ is called *fact*, and the set of all facts is denoted by $F$. A *partial assignment* $p$ maps a subset of variables $vars(p) \subseteq \mathcal{V}$ to values in their domains. For a variable $v \in \mathcal{V}$ and a partial assignment $p$, the value of $v$ in $p$ is denoted by $p[v]$ if $v \in vars(p)$ and we say $p[v]$ is *undefined* otherwise. A full assignment $s$ is called a *state*, and the set of all states is denoted by $\mathcal{S}$. State $s$ is *consistent* with a partial assignment $p$ if they agree on all variables in $vars(p)$, denoted by $p \subseteq s$. Each action $o$ in $\mathcal{O}$ is a pair $\langle pre(o), eff(o) \rangle$, where $pre(o)$ and $eff(o)$ are partial assignments called *precondition* and *effect*, respectively. Furthermore, $o$ has an associated non-negative cost denoted by $C(o) \in \mathbb{R}^{0+}$. An action $o$ is applicable in state $s$ if $pre(o) \subseteq s$. Applying $o$ in $s$ results in a state denoted by $s[\![o]\!]$, where $s[\![o]\!][v] = eff(o)[v]$ for all $v \in vars(eff)$, and $s[\![o]\!][v] = s[v]$ for all other variables. An action sequence $\pi = \langle o_1, \ldots, o_n \rangle$ is applicable in state $s$ if there are states $\langle s_0, \ldots, s_n \rangle$ such that $o_i$ is applicable in $s_{i-1}$ and $s_{i-1}[\![o_i]\!] = s_i$ for $0 \leq i \leq n$. We denote $s_n$ by $s[\![\pi]\!]$. An action sequence with $s_\star \subseteq s_0[\![\pi]\!]$ is called a *plan*. The cost of a plan $\pi$, denoted by $C(\pi)$ is the sum of the costs of the actions in the plan. The set of all plans is denoted by $\mathcal{P}_\Pi$, and an *optimal* plan is a plan in $\mathcal{P}_\Pi$ with the minimum cost. Next, we present the *top-$k$ planning problem*, as defined by Sohrabi et al.; Katz et al. (2016; 2018).

**Definition 1 (top-$k$ planning problem)** *Given a classical planning task $\Pi$ and a natural number $k$,* top-$k$ *planning problem is the task of finding a set of plans $P \subseteq \mathcal{P}_\Pi$ that satisfy the following properties:*

- *For all plans $\pi \in P$, if there exists a plan $\pi' \in \mathcal{P}_\Pi$ such that $C(\pi') < C(\pi)$, then $\pi' \in P$,*
- *$|P| \leq k$, and if $|P| < k$, then $P = \mathcal{P}_\Pi$.*

We say a top-$k$ planning problem $\langle \Pi, k \rangle$ is solvable if $|P| = k$ and unsolvable if $|P| < k$. Note that cost-optimal planning is a special case of top-$k$ planning for $k = 1$.

**Single Goal State Reformulation**   One of the limitations of $K^*$ that has not been discussed in the planning literature is its restriction to graphs with a single terminal node. In classical planning, tasks can have many goal states if the goal $s_\star$ is a partial assignment. In such cases, we can reformulate $\Pi$ into a planning task with a single goal state. Katz et al. (2018) showed such a reformulated task as $\Pi^g = \langle \mathcal{V}^g, \mathcal{O}^g, s_0^g, s_\star^g \rangle$, where $\mathcal{V}^g = \mathcal{V} \cup \{v_g\}$ with a binary indicator variable $v_g$ for reaching a goal state, $\mathcal{O}^g = \{\langle pre(o) \cup \langle v_g, 0 \rangle, eff(o) \rangle | o \in \mathcal{O}\} \cup \{o_g\}$ with a goal-achieving action $o_g$ such that $pre(o_g) = s_\star \cup \{\langle v_g, 0 \rangle\}$ and $eff(o_g) = \{\langle v_i, t[v_i] \rangle | v_i \in vars(t)\} \cup \{\langle v_g, 1 \rangle\}$ for an arbitrary full state $t$, $s_0^g = s_0 \cup \{\langle v_g, 0 \rangle\}$, and $s_\star^g = eff(o_g)$.

In words, the auxiliary goal-achieving zero-cost action can be applied only once upon reaching the original goal, changing the state to the new goal state. After that, no action is applicable in the new goal state, and therefore there is a one-to-one correspondence between the plans of $\Pi$ and those of $\Pi^g$. It is worth noting that such task transformations indeed impact the quality of domain-independent heuristics. In the rest of the paper, we will assume that all planning tasks are reformulated as a single goal-state task and drop superscript $g$ in the notation if it is clear from the context.

### Eppstein's Algorithm and $K^*$ Search

We will review high-level ideas of $EA$ in $K^*$ search, especially on the heap-based data structures and criteria for safely enumerating the $k$ best paths from partially explored search graph using $A^*$. For more details, please refer to Eppstein (1998) and Aljazzar and Leue (2011).

**Eppstein's Implicit Path Representation**   We assume familiarity with $A^*$, and begin by introducing notations related to $EA$ in the context of $K^*$ search. Denoting the explicit search graph explored by $A^*$ as $G_{A^*}$, $EA$ can utilize the shortest path tree $T_{A^*}$ maintained by $A^*$ to represent all paths between the initial state $s_0$ and the unique goal state $s_\star$ using a sequence of "side-tracked" edges (STE), where each STE $(u, v) \in G_{A^*} \setminus T_{A^*}$. For each STE $(u, v)$ between nodes from $u$ to $v$, we can compute the deviation cost $\delta_o(u, v)$ through an action $o$ against the cost of the incoming edge toward $v$ in $T_{A^*}$ by $\delta_o(u, v) = g(u) + C(o) - g(v)$ if and only if $v = u[\![o]\!]$. We denote an arbitrary goal reaching path from $s_0$ in $G_{A^*}$ by $\rho_{A^*}(s_0, s_\star)$, and the unique path from $u$ to $v$ in $T_{A^*}$ by $\rho_{A^*}^*(u, v)$ if it exists. Then, any $\rho_{A^*}(s_0, s_\star)$ can be uniquely represented by an ordered sequence of STEs, denoted by $\text{SIDETRACKS}(\rho_{A^*}(s_0, s_\star)) = \langle (u_1, v_1), \ldots, (u_q, v_q) \rangle$, where an STE closer to $s_\star$ appears earlier in the sequence. Namely, $\rho_{A^*}(s_0, s_\star)$ can be reconstructed from $\text{SIDETRACKS}(\rho_{A^*}(s_0, s_\star))$ by

$$\rho_{A^*}(s_0, s_\star) = \rho_{A^*}^*(s_0, u_q) \circ [\circ_{i=q}^2 \{(u_i, v_i) \circ \rho_{A^*}^*(v_i, u_{i-1})\}]$$
$$\circ (u_1, v_1) \circ \rho_{A^*}^*(v_1, s_\star),$$

where ∘ concatenates edges and paths from left to right.

**Heaps for Representing Paths** $EA$ traverses a search graph called path graph $G_{EA}$, where each node encodes a path $\rho_{A^*}(s_0, s_\star)$. In order to generate $G_{EA}$, $EA$ maintains the following three heap-based data structures:

- *incoming heap* $H_{\text{in}}(v)$, a binary heap for storing all incoming STEs $(\cdot, v)$ toward $v \in T_{A^*}$ sorted by $\delta(\cdot, v)$,
- *tree heap* $H_{\text{T}}(t)$, a binary heap for storing the root node ROOTSTE$(v)$ of $H_{\text{in}}(v)$ for every node $v$ in the shortest path $\rho_{A^*}^*(s_0, t)$ from $s_0$ to $t$ in $T_{A^*}$,
- *graph heap* $H_{\text{G}}(t)$, a 3-heap that merges $H_{\text{T}}(t)$ and $H_{\text{in}}(v)$ for all $v \in \rho_{A^*}^*(s_0, t)$.

We can update all *incoming heaps* while generating search nodes in $G_{A^*}$ and maintain all STEs sorted with respect to the deviation cost. However, *tree heaps* and *graph heaps* should be created after all STEs in $G_{A^*}$ have been seen. By merging a *tree heap* and *incoming heaps*, $H_{\text{G}}(t)$ can enumerate all paths with a single deviation relative to $\rho_{A^*}^*(s_0, t)$ sorted by the deviation cost. To be concrete, recall that $H_{\text{G}}(t)$ merges ROOTSTEs that are stored in $H_{\text{T}}(t)$, and other STEs that are descendants of ROOTSTE$(v)$ in $H_{\text{in}}(v)$. Therefore, there are two types of nodes in the *graph heap*. Denoting the root node of the heap maintained by $H_{\text{G}}(t)$ by $R(t)$, the first node traversed from the root $R(t)$ must be a ROOTSTE$(v)$. From a node in $H_{\text{G}}(t)$ that originated from $H_{\text{T}}(t)$, there are two possible cases for generating children nodes. In the first case, the child node is one of the two children nodes in $H_{\text{T}}(t)$ because it is a binary heap. In the second case, the child node is the only child of ROOTSTE$(v)$ in $H_{\text{in}}(v)$ because $H_{\text{in}}(v)$ in $EA$ stores one child under ROOTSTE$(v)$. Next, from a node that originated from *incoming heaps* $H_{\text{in}}(v)$, there is only a single case for generating successor nodes that are, at most, two children in the binary heap $H_{\text{in}}(v)$. We have illustrated how $H_{\text{G}}(t)$ expands nodes associated with STEs in a 3-heap, which is the basis for generating $G_{EA}$.

**Path Graph** By extending $H_{\text{G}}(t)$ to enumerate all paths $\rho_{A^*}(s_0, t)$ with a single deviation relative to $\rho_{A^*}^*(s_0, t)$, $G_{EA}$ expands nodes in the search graph that can enumerate all SIDETRACKS$(\rho_{A^*}(s_0, s_\star))$. In the following, we will illustrate how the successor generator of $G_{EA}$ works. Without loss of generality, each node $n_{(u_q, v_q)}$ in $G_{EA}$ associated with the STE $(u_q, v_q)$ stores a pointer to the node in some *graph heap* $H_{\text{G}}(t)$ that stores $(u_q, v_q)$, and it also tracks a sequence of STEs, $\langle (u_1, v_1), \ldots, (u_{q-1}, v_{q-1}) \rangle$ selected for the deviation, which will become clear in a moment. The successor generator of $G_{EA}$ makes two types of decisions encoded by two types of edges:

- a *cross-edge* that extends SIDETRACKS$(\rho_{A^*}(s_0, s_\star))$ to a sequence of STEs $\langle (u_1, v_1), \ldots, (u_q, v_q) \rangle$,
- a *heap-edge* that only generates children from the associated node in some *graph heap* $H_{\text{G}}(t)$.

Starting from a special root node $\mathcal{R}$ of $G_{EA}$ associated with an empty sequence of SIDETRACKS$(\rho_{A^*}^*(s_0, s_\star))$, the successor generator generates a single child node associated with $R(s_\star)$, which is the root node of $H_{\text{G}}(s_\star)$ that stores the

**Algorithm 1** Generic $K^*$ Search for Top-$k$ Planning

---
**Input:** Single goal-state planning task $\Pi$, $k$
**Output:** Top-$k$ plans
1: Initialize $K^*$ search
2: $P \leftarrow \emptyset$             ▷ Initialize set of found plans
3: **while** True **do**             ▷ $K^*$ outer-loop
4:      **while** $\neg\big(\text{OPEN}_{A^*} = \emptyset \vee \text{SWITCH-TO-}EA(\ )\big)$ **do**
5:          Expand an $A^*$ node
6:          Update data structures $G_{A^*}$, $T_{A^*}$, $H_{\text{in}}$, etc
7:      PREPARE$EA$()      ▷ Update $H_{\text{T}}$, $H_{\text{G}}$, $G_{EA}$, etc
8:      **while** $\neg\big(\text{OPEN}_{EA} = \emptyset \vee \text{SWITCH-TO-}A^*(\ )\big)$ **do**
9:          Expand a node in path graph $G_{EA}$
10:          Reconstruct a plan and update $P$
11:          **if** $|P| = k$ **then return** $P$
12:      **if** $\text{OPEN}_{A^*}$ and $\text{OPEN}_{EA}$ are empty **then return** $P$

---

STE $(u_1, v_1)$. Next, the node $R(s_\star)$ generates its successors following two possible cases. The first case follows *heap-edges* that are children of $R(s_\star)$ in $H_{\text{G}}(s_\star)$ by deferring a decision for making a deviation. The second case follows a single cross-edge which creates a child node pointing to the root node $R(u_1)$ of $H_{\text{G}}(u_1)$ with an extended STE sequence $\langle (u_1, v_1) \rangle$. The rest of the nodes in $G_{EA}$ are generated as illustrated above, namely, generating successor nodes following *heap-edges* in *graph heaps* or starting to enumerate the next possible deviations relative to $\rho_{A^*}^*(s_0, u_q)$ when $(u_q, v_q)$ was the most recently added STE to the sequence. The maximum branching factor in $G_{EA}$ is at most 4 since an additional *cross-edge* that doesn't exist in $H_{\text{G}}(v)$ is added. Since each node in $G_{EA}$ extends SIDETRACKS$(\rho_{A^*}(s_0, s_\star))$ in increasing order of the total cost, expanding $G_{EA}$ and running Dijkstra's algorithm (Dijkstra 1959) on $G_{EA}$ will enumerate all shortest paths in increasing order of path costs.

**Generic $K^*$ Search and Existing Variants** Algorithm 1 shows a generic template for adapting $K^*$ search for top-$k$ planning. Following the template, we can divide generic $K^*$ search into three parts:

- $A^*$ with updates on heaps and checking the switching criteria from $A^*$ to $EA$ (line 4–6),
- PREPARE$EA$() for updating heaps and other data structures (line 7).
- $EA$ that extracts solution from $G_{A^*}$ and checks switching criteria from $EA$ to $A^*$ (line 8–11),

The main contribution of the original $K^*$ is to show a safe switching criterion from $EA$ to $A^*$ that ensures the soundness of $EA$ when it processes a partially explored search graph $G_{A^*}$ with admissible heuristics. Specifically, after expanding $G_{A^*}$, the original $K^*$ reconstructs a solution path from the expanded nodes in $G_{EA}$ only if the maximum path cost of the children nodes of the head node in $\text{OPEN}_{EA}$ is not greater than the head value of $\text{OPEN}_{A^*}$.

Unfortunately, the original $K^*$ algorithm did not provide details about the implementation of PREPARE$EA$(). When $A^*$ uses consistent heuristics, $\text{OPEN}_{A^*}$ expands nodes in the order of the path cost, and $T_{A^*}$ monotonically adds a

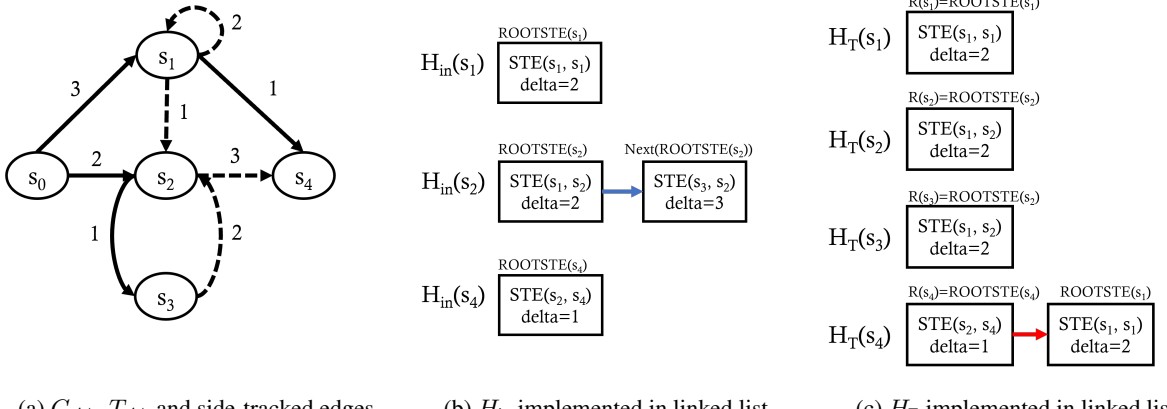

(a) $G_{A^*}$, $T_{A^*}$ and side-tracked edges  (b) $H_{\text{in}}$ implemented in linked list  (c) $H_{\text{T}}$ implemented in linked list

Figure 1: Illustration of *incoming heaps* $H_{\text{in}}(v)$ and *tree heaps* $H_{\text{T}}(v)$ implemented as linked lists. Figure 1a reproduces the example shown in the original $K^*$ paper by (Aljazzar and Leue 2011), where the source node is $s_0$ and the terminal node is $s_4$. The solid edges represent the search tree $T_{A^*}$, and the dotted edges are sided-tracked edges. The cost of each edge appears next to it. Figure 1b shows the *incoming heaps* for each node in $G_{A^*}$, except for $H_{\text{in}}(s_0)$ and $H_{\text{in}}(s_3)$, which are empty. Each node in the linked list stores a pointer to a data structure that stores the STE $(u, v)$ and its deviation cost. Figure 1c shows the *tree heaps* for each node in $G_{A^*}$, except for node $s_0$ because there is no path from $s_0$ to $s_0$. Each $H_{\text{T}}(v)$ stores all ROOTSTEs over the path from $s_0$ to $v$ in $T_{A^*}$. For example, $H_{\text{T}}(s_4)$ collects and sorts ROOTSTE$(s_1)$ in $H_{\text{in}}(s_1)$ and ROOTSTE$(s_4)$ in $H_{\text{in}}(s_4)$.

new edge when $A^*$ expands a new state. This implies that all heaps required for $EA$ can be reused without the need for PREPARE$EA()$, as illustrated in the original $K^*$ algorithm. However, $A^*$ does not ensure such a monotonic behavior for generated nodes, and we need to carefully sort *incoming heaps* and properly update *tree heaps* and *graph heaps* if we want to apply $EA$ on the nodes generated from $A^*$. When inconsistent heuristics guide $A^*$, the original $K^*$ suggested refreshing the path graph to make it consistent with $G_{A^*}$. However, due to the missing details, existing variants of $K^*$ for top-$k$ planning couldn't incorporate inconsistent heuristics. The following section will present details of PREPARE$EA()$.

The switching criterion SWITCH-TO-$EA()$ is less crucial than the other two parts. The original $K^*$ proposed to escape $A^*$ if the number of nodes newly expanded exceeds 20 percent of the total number of nodes in OPEN$_{A^*}$ counted at the previous iteration. Existing variants also follow this ad-hoc rule.

## Renovating $K^*$ Search for Top-$k$ Planning

We have been motivated to renovate $K^*$ search for top-$k$ planning by observing the discrepancy between the strong theoretical guarantee and the weak empirical evaluation results reported up to date. Among the handful of available open-source implementations, for example, (Shibuya and Imai 1997; Katz et al. 2018), none has implemented 3-heaps with non-standard heap operations for pointing and traversing nodes inside the heap, to the best of our knowledge. While reviewing the high-level ideas in $EA$, we saw that heap-based data structures, including the path graph, share nodes that store STEs across multiple heaps, and they may redirect nodes if necessary. To manage these heap nodes more flexibly, we implemented alternative data structures

that use a linked list, which demonstrated satisfactory performance in our evaluation. In addition to improving the implementation of $EA$, we also identified several pitfalls that need to be overcome to achieve promising performance of $EA$ in the generic $K^*$ search template. In particular, these include missing details on PREPARE$EA()$ and more aggressive yet safe switching criteria between $A^*$ and $EA$, which will be revisited in each part of the generic $K^*$. Next, we will begin by addressing the restriction to tasks with a single goal state, an uncommon situation in classical planning.

## Preserving Heuristic Quality after Reformulation

As discussed, previous approaches chose to transform the planning task to an equivalent (in terms of plan spaces) planning task with a single goal state and solve that task instead. As a result, heuristic functions were computed on the transformed task and might have resulted in less informative estimates. Note that while the search requires this limitation to a single goal state, the heuristics do not, and it is possible to transform the task back to the original task. Inspired by the idea of Domshlak, Katz, and Lefler (2012), we propose to perform the search on the transformed single goal state task but evaluate it on the task transformed back to original. The following theorem shows the validity of such an approach.

**Theorem 1** *Let $\Pi$ be a planning task and $\Pi^g$ be its single-goal transformation. Let $m$ be a mapping from the states of $\Pi^g$ to the states of $\Pi$ obtained by projecting away the additional variable $v_g$. If $h$ is an admissible heuristic function for $\Pi$, then $h_g$ defined by $h_g(s) = 0$ if $s$ is the goal state of $\Pi^g$, and otherwise $h_g(s) = h(m(s))$, is an admissible heuristic function for $\Pi^g$. Moreover, If $h$ is consistent, then $h_g$ is also consistent.*

**Proof:** For each non-goal state $s$ in $\Pi^g$, an optimal plan $\pi_g$ for $s$ can be mapped to an optimal plan $\pi$ for $m(s)$ in

$\Pi$ by simply removing the last zero-cost action $o_g$. Since $h(m(s)) \leq C(\pi) = C(\pi_g)$, we have $h_g(s) = h(m(s)) \leq C(\pi_g)$. Let $t$ be a non-goal state obtained from $s$ by applying $o$ in $\Pi^g$. Then $m(t)$ is obtained from $m(s)$ by applying $o$ in $\Pi$ and since $h$ is consistent, we have $h(m(s)) \leq h(m(t)) + C(o)$ and therefore $h_g(s) \leq h_g(t) + C(o)$. If $t$ is the goal state, it is obtained from $s$ by applying $o_g$. The precondition of $o_g$ includes the goal of $\Pi$ and thus we have $m(s)$ being a goal state for $\Pi$, meaning $h_g(s) = h(m(s)) = 0$. In this case as well we have $0 = h_g(s) \leq h_g(t) + C(o_g) = 0$. $\square$

### List-based Implementation of $EA$

We follow the high-level idea for implementing $EA$ as shown in the background, yet introducing two changes.

**List-based $H_{\mathbf{in}}$ and $H_{\mathbf{T}}$**  We simplify the heap-based data structures for generating $G_{EA}$ by building $H_{\text{in}}(v)$ and $H_{\text{T}}(v)$ using linked lists instead. We skip the step of merging them to create *graph heaps* $H_{\text{G}}(v)$. To illustrate changes, Figure 1 reproduces the same example shown in (Aljazzar and Leue 2011). Figure 1b and 1c show a simpler variation of $H_{\text{in}}(v)$ and $H_{\text{T}}(v)$. Note that every node in the linked list only stores a pointer to the data structure storing STEs, and no duplicate STE $(u, v)$ will be created. We admit that the presented approach loosens the complexity bound because inserting a node to *incoming heaps* and *tree heaps* implemented in linked lists takes linear time. However, we see that this simplification didn't degrade the overall performance while solving planning tasks, since the branching factor of the search graph of planning tasks is usually much smaller than the total number of nodes in $G_{A^*}$. For denser graphs, the ideal way would be fully implement heaps with pointer-friendly data structures instead of arrays or vectors.

**Path Tree with a List-based Successor Generator**  As *graph heaps* are no longer built explicitly, we implemented a different successor generator for exploring $G_{EA}$. The new successor generator traverses the nodes in the linked lists implementing $H_T$ and $H_{in}$. Utilizing the fact that $H_{in}(v)$ and $H_T(v)$ are implemented as linear linked lists, a path graph node $n_{EA}$ stores a tuple $\langle loc(H_T(v)), loc(H_{in}(v)), pa, value, crossing \rangle$ for enumerating the $k$-best solutions, where $loc$ refers to the location of a linked list node in either $H_T$ or $H_{in}$ that stores a pointer to STE, $pa$ is a pointer to its parent path graph node, $value$ is the path cost value, and $crossing$ is a Boolean flag indicating whether the node follows a crossing-arc. Since all linked lists are sorted by the deviation cost and $H_T(t)$ collects only ROOTSTE$(v)$ of $H_{in}(v)$ for $v \in \rho_{A^*}^*(s_0, t)$, a path graph node $n_{EA}$ can generate at most 3 children nodes:

- a node associated with the next node in $H_T(v)$,
- a node associated with the next node in $H_{in}(v)$, and
- a node that follows a *cross-edge*.

Note that the branching factor is at most 3 because linked lists are linear.

Figure 2 illustrates a path tree $G_{EA}$ generated from $G_{A^*}$, shown in Figure 1a. The root of $G_{EA}$ stores the location of the root node $R(s_4)$ of the linked list $H_{\text{T}}(s_4)$, since $R(s_4)$

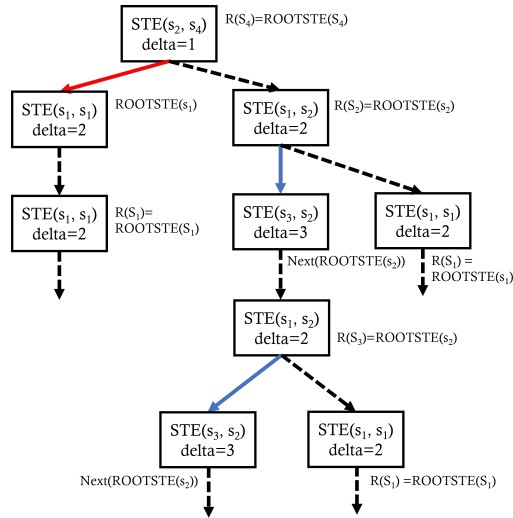

Figure 2: Path tree $G_{EA}$ generated from $G_{A^*}$ in Figure 1a. Each node represents a path graph node, and there are three types of arrows: a dotted arrow, a solid red arrow, and a solid blue arrow. The dotted arrow represents a *cross-edge*, the solid red arrow represents a *heap-edge* in $H_T$, and the solid blue arrow represents a *heap-edge* in $H_{\text{in}}$.

points to an STE with the minimum deviation from the optimal solution path from $s_0$ to $s_4$. Following a *heap-edge*, a child path graph node can point to either the next ROOTSTE in $H_{\text{T}}(s_4)$ or the next STE in $H_{\text{in}}(s_4)$. We see that $H_{\text{in}}(s_4)$ doesn't have the next node shown in Figure 1b, whereas $H_{\text{T}}(s_4)$ has the next node ROOTSTE$(s_1)$ in Figure 1c, following a red *heap-edge*. Therefore, we generate a single child following the red *heap-edge* that encodes a possible deviation $(s_1, s_1)$ relative to $\rho_{A^*}^*(s_0, s_4)$. Following a *cross-edge* in a dotted arrow from the root path graph node that stores the location of $R(s_4)$, we generate a child path graph node that points to $R(s_2)$. We emphasize that a *cross-edge* represents the decision to deviate from $T_{A^*}$. Therefore, the successor generator chooses to deviate on $(s_2, s_4)$ with a deviation cost of 1. The next possible deviation can occur between $s_0$ and a node in $\rho_{A^*}^*(s_0, s_2)$. As a result, the child path graph node points to $R(s_2)$ in $H_{\text{T}}(s_2)$. The successor generator generates all other nodes in the same manner. However, unlike the path graph that connects all the *graph heaps* $H_G$ and results in a search graph, our variant generates a path tree.

We have presented a simple variation for $EA$ that generates search nodes by traversing the links in the linked lists $H_T$ and $H_{in}$ that sort STEs, resulting in a search tree. For the rest of $EA$, Dijkstra's algorithm enumerates shortest paths. Due to the modifications to $EA$ implementation, our variant performs tree search and sorts STEs more often than the original heap-based approach. However, we didn't observe any notable degradation in search performance while solving top-$k$ planning problems, as the number of requested plans is relatively low, for example, less than 10,000, compared to the total number of nodes in $G_{A^*}$.

**Algorithm 2** Switch from $EA$ to $A^*$

**Input:** Flag rs-$EA$ indicating whether to restart $EA$
1: **function** SWITCH-TO-$A^*$(rs-$EA$)
2:     $n_{\text{head}} \leftarrow \text{top}(\text{OPEN}_{EA})$
3:     **if** rs-$EA$ **then** $EA_{Thr} \leftarrow n_{\text{head}}.\text{value}$
4:     **else** $EA_{Thr} \leftarrow \max \{ \text{children}(n_{\text{head}}).\text{value} \}$
5:     **return** $EA_{Thr} > f_{min}$

## Revisiting $EA$ inside $K^*$

If we examine the safe switching criteria of the original $K^*$, we see that $EA$ has to defer extracting valid top-$k$ plans when its child has path cost greater than $f_{\min}$ value, the current minimum value of $\text{OPEN}_{A^*}$. This choice guarantees that the successor generator of $G_{EA}$ won't skip any solution path with a path cost lying between the costs of the two path graph nodes, the head node in $\text{OPEN}_{EA}$ and its child nodes, when new STEs are added in $A^*$. Namely, if the heaps maintain the insertion order for the STEs with the same deviation cost, $K^*$ can skip refreshing heap-based data structures. However, $A^*$ may need to expand significant number of nodes to reach the next additional layer during $A^*$. Furthermore, if a guiding heuristic function is inconsistent, $K^*$ must purge and rebuild the heaps anyway, losing the benefit of keeping the safe switching criteria.

Based on this observation, we present a new variant with more aggressive, yet still safe switching criteria in Algorithm 2 that restarts $EA$ at every outer iteration. SWITCH-TO-$EA$(rs-$EA$) returns whether $EA_{Thr}$ is greater than $f_{\min}$ or not, where $EA_{Thr}$ is taken from the head value of $\text{OPEN}_{EA}$ if rs-$EA$ was True (line 3), or the same value as the original $K^*$ otherwise (line 4). Note that $K^*$ utilizes $EA_{Thr}$ when it switches from $A^*$ to $EA$ in Algoroithm 4.

## Revisiting PREPARE$EA$()

Next, we revisit PREPARE$EA$() in Algorithm 3 when $A^*$ uses either consistent or inconsistent heuristics. We first consider consistent heuristics. Although the expanded nodes in $G_{A^*}$ appear in the order of their path cost, and $T_{A^*}$ only adds edges as $A^*$ continues its search, we cannot assume such a monotonic behavior for nodes generated during $A^*$ and, hence, STEs seen while generating these nodes. Namely, if we perform $EA$ search over the path graph nodes associated with an STE $(u, v)$ with $v$ not yet expanded in $A^*$, we must ensure that the heaps are well sorted in linked lists. In addition, if we don't restart $EA$ at every $K^*$ iteration, we must ensure that the heaps are stable (preserve the order of

**Algorithm 3** Prepare $EA$

**Input:** Flag rs-$EA$ indicating whether to restart $EA$
1: **function** PREPARE$EA$(rs-$EA$)
2:     **if** rs-$EA$ **then** Clear $H_T$ and $\text{OPEN}_{EA}$
3:     **if** reopen occurred **then** Rebuild $H_{\text{in}}$, Recompute $\delta$
4:     **if** $\text{OPEN}_{EA}$ is not empty **then return**
5:     **if** $\text{OPEN}_{A^*}$ is empty $\vee Root(G_{EA}).\text{value} <= f_{min}$
      **then** Push $Root(G_{EA})$ to $\text{OPEN}_{EA}$

**Algorithm 4** Switch from $A^*$ to $EA$

**Input:** Flag sog, indicating whether to switch on generating a goal, thresholds $A_{lb}^*$ and $A_{ub}^*$
1: **function** SWITCH-TO-$EA$(sog, $A_{lb}^*$, $A_{ub}^*$)
2:     **if** sog $\wedge$ goal node generated **then return** True
3:     **if** $A_{iter}^* \geq A_{ub}^*$ **then return** True
4:     **if** $EA_{Thr}$ unknown $\wedge A_{iter}^* > A_{lb}^*$ **then return** True
5:     **if** $\neg$reopen **then return** $EA_{Thr} \leq f_{min}$
6:     **else if** $(A_{iter}^* > A_{lb}^*)$ **return** $EA_{Thr} \leq f_{min}$

insertion for tied items), so as not to miss a child during the expansion of $G_{EA}$. In the new variant that restarts $EA$ every iteration, PREPARE$EA$(rs-$EA$) clears $H_T(v)$ and $\text{OPEN}_{EA}$ (line 2) before restarting $EA$. When a heuristic function is inconsistent, the behavior is the same as consistent heuristics as long as $A^*$ does not reopen any node. However, when $A^*$ reopens a node, the value of all descendant nodes become invalidated and we must recompute all the deviation costs for those nodes. Instead of checking the validity of the deviation costs after a reopen, PREPARE$EA$(rs-$EA$) rebuilds $H_{\text{in}}(v)$ (line 3). After clearing any possibly invalidated $H_T$ and $\text{OPEN}_{EA}$ due to the aggressive switching criteria shown in Algorithm 2, PREPARE$EA$(rs-$EA$) pushes the root node $Root(G_{EA})$ to $\text{OPEN}_{EA}$ to initiate $EA$ if either $A^*$ queue is exhausted or the second suboptimal solution is safe to extract in $EA$ (line 5).

## Revisiting $A^*$ inside $K^*$

Expanding $A^*$ nodes follows the standard $A^*$ algorithm. The only modification needed is updating $H_{\text{in}}(v)$, which stores STEs $(\cdot, v)$, on generating $A^*$ nodes. In the earlier approach by Riabov, Sohrabi, and Udrea (2014), $G_{A^*}$ and other heap-based data structures were created after escaping $A^*$. In the new variant, we maintain a set $S_{\text{in}}(v)$ for each $H_{\text{in}}(v)$, and each $S_{\text{in}}(v)$ stores all $(\cdot, v)$s on generating nodes in $A^*$. Since $A^*$ updates $T_{A^*}$ with expanded nodes, we defer the creation of $H_{\text{in}}(v)$ until $v$ is expanded. After updating $T_{A^*}$ with expanded $v$, we add all $(\cdot, v)$s from $S_{\text{in}}(v)$ to $H_{\text{in}}(v)$. If a *reopen* occurs during $A^*$, we only add $(\cdot, v)$s to $S_{\text{in}}(v)$ since PREPARE$EA$(rs-$EA$) will rebuild $H_{\text{in}}(v)$ after escaping $A^*$.

Next, we revisit the switching criteria from $A^*$ to $EA$. Instead of following the earlier choice that relied on a single threshold on the number of $A^*$ nodes expanded, denoted as $A_{iter}^*$, we introduce a lower bound $A_{lb}^*$ and an upper bound $A_{ub}^*$, as well as an additional Boolean flag sog, indicating whether to switch on generating a goal. Algorithm 4 shows that it will escape $A^*$ if a goal node was generated and sog was set to $True$ (line 2), or if $A^*$ expanded nodes up to the upper bound $A_{ub}^*$ (line 3). Otherwise, it will expand $A_{lb}^*$ nodes for the first $A^*$ iteration after finding the shortest path (line 4). If *reopen* did not occur during $A^*$, we switch to $EA$ if $EA_{Thr} \leq f_{min}$ (line 5). If $A^*$ reopened a node, we use the same criteria but expand at least $A_{lb}^*$ (line 6).

## Algorithm 5 A New $K^*$ Search for Top-$k$ Planning

**Input:** Reformulated planning task $\Pi$, number of plans $k$, flag indicating whether to restart $EA$ rs-$EA$, flag indicating whether to switch on goal generation sog, bounds on the number of $A^*$ expansions $A^*_{lb}$, $A^*_{ub}$
**Output:** Top-$k$ solutions
1: Initialize $A^*$ search
2: **while** True **do**
3:     **while** $\neg\big($OPEN$_{A^*}$ is empty $\vee$
4:         SWITCH-TO-$EA$(sog, $A^*_{lb}$, $A^*_{ub}$)$\big)$ **do**
5:        Expand an $A^*$ node $u$
6:        Increase $A^*_{iter}$
7:        **For each** node $v$ generated from u
8:           Add STE $(u, v)$ to set $S_{in}(v)$
9:           **if** $\neg$ reopen occurred **then**
10:             Insert $(u, v)$ to $H_{in}(v)$, Sort $H_{in}(v)$
11:        **if** reopen occurred **then** $A^*_{iter} \leftarrow 0$
12:     PREPARE$EA$(rs-$EA$)
13:     **while** $\neg\big($OPEN$_{EA}$ is empty $\vee$
14:         SWITCH-TO-$A^*$(rs-$EA$) $\big)$ **do**
15:        Expand an $EA$ node $n_{EA}$
16:        Lazy build $H_T$ while expanding $G_{EA}$
17:        Reconstruct a plan from $n_{EA}$
18:        **if** k plans found **then return** Top-$k$ plans
19:     **if** OPEN$_{A^*}$ is empty $\wedge$ OPEN$_{EA}$ is empty **then**
20:        **return** Found plans

## A New $K^*$ Search for Top-$k$ Planning

Algorithm 5 presents a new variant of the generic $K^*$ algorithm that integrates three revisited parts. We will highlight the modification made to the existing $K^*$ algorithm. The input parameters are the number of requested plans $k$, a Boolean flag rs-$EA$ to restart $EA$ after switching from $A^*$, a lower bound $A^*_{lb}$ and an upper bound $A^*_{ub}$ on the number of node expansions in $A^*$, and a Boolean flag sog to switch from $A^*$ to $EA$ upon generating a goal node in $A^*$.

The $K^*$ search starts with initializing an open list OPEN$_{A^*}$ and a closed list CLOSED$_{A^*}$ for $A^*$ algorithm (line 1). Then, $A^*$ expands nodes until either OPEN$_{A^*}$ becomes empty or the switching criteria from $A^*$ to $EA$ are satisfied (lines 3–11). While generating nodes during $A^*$, we insert each STE $(u, v)$ to a set $S_{in}(v)$ associated with $v$ (line 8) and maintain $H_{in}(u)$ sorted if state $u$ was expanded (line 9–10). We reset the counter for the number of expanded nodes, $A^*_{iter}$, to zero if $A^*$ repones a node (line 11). After escaping $A^*$, PREPARE$EA$(rs-$EA$) prepares $EA$ (lines 12). $EA$ explores $G_{EA}$ until either OPEN$_{EA}$ becomes empty or the switching criteria are satisfied (lines 13–18). While explicating the path tree $G_{EA}$, we lazy build $H_T$ since it will be cleared up in PREPARE$EA$(rs-$EA$) when rs-$EA$ is True (line 15). SWITCH-TO-$EA$(rs-$EA$) updates $EA_{Thr}$ and ensures that a valid plan can be reconstructed from the expanded path graph node $n_{EA}$ (line 17). If $EA$ finds $k$ requested plans, then $EA$ returns the top-$k$ plans (line 18). Finally, if both OPEN$_{A^*}$ and OPEN$_{EA}$ become empty before finding $k$ plans, $K^*$ proves that the problem is unsolvable

|  | (1) | (2) | (3) | (4) | (5) | (6) | (7) | (8) | (9) | (10) | (11) |
|---|---|---|---|---|---|---|---|---|---|---|---|
| FI (1) | 0 | 18 | 18 | 16 | 23 | 20 | **30** | **31** | 26 | 29 | 27 |
| SymK (2) | **37** | 0 | **30** | **28** | **30** | 30 | **41** | **40** | **36** | **37** | 34 |
| LMcut↓(3) | **37** | 24 | 0 | 0 | 24 | 16 | **35** | 39 | 25 | 36 | 32 |
| LMcut(4) | **39** | 25 | 14 | 0 | 26 | 19 | 37 | **40** | 28 | 37 | 33 |
| iPDB↓(5) | **35** | 28 | 18 | 15 | 0 | 4 | 34 | 38 | 26 | 38 | 31 |
| iPDB(6) | **40** | 30 | 27 | 22 | 17 | 0 | 34 | **40** | 31 | 42 | 35 |
| prev-iPDB↓(7) | 28 | 18 | 11 | 8 | 11 | 9 | 0 | 25 | 13 | 22 | 19 |
| CEGAR↓(8) | 26 | 15 | 5 | 5 | 7 | 6 | 18 | 0 | 2 | 17 | 16 |
| CEGAR(9) | 32 | 19 | 15 | 10 | 16 | 11 | **26** | **28** | 0 | 24 | 20 |
| M&S↓(10) | 29 | 17 | 10 | 10 | 7 | 6 | **29** | 26 | 20 | 0 | 8 |
| M&S(11) | 31 | 16 | 16 | 15 | 12 | 11 | **32** | 26 | 22 | 17 | 0 |
| Total | 633 | 892 | 895 | **921** | 805 | 852 | 670 | 721 | 799 | 720 | 776 |

Table 1: Domain-level performance, comparing the single goal task (denoted by adding ↓ to the heuristic name) to task transformation back to original, for $k = 1000$ and four different heuristics. The last row depicts the overall coverage.

and returns the found plans (line 19–20).

## Experimental Evaluation

To empirically compare the suggested algorithm, we have implemented our variant of $K^*$ on top of the Fast Downward planning system (Helmert 2006a). All experiments were performed on Intel(R) Xeon(R) Gold 6248 CPU @ 2.50GHz machines, with the timeout of 30 minutes and memory limit of 8GB per run. The benchmark set consists of all benchmarks from optimal tracks of International Planning Competitions 1998-2018, a total of 1827 tasks in 65 domains. We have experimented with four admissible heuristics, LMcut (Helmert and Domshlak 2009), merge-and-shrink abstraction (denoted by M&S) (Helmert, Haslum, and Hoffmann 2007), counterexample-guided Cartesian abstraction refinement (denoted by CEGAR) (Seipp and Helmert 2018), and pattern database heuristic iPDB (Haslum et al. 2007). While the latter three are also consistent, LMcut is not. We measure the total time for finding the top-k solution and the coverage: 1 per task if the top-k solution was found, 0 otherwise.

We start by comparing evaluating the states on the single goal transformation as in previous work (Riabov, Sohrabi, and Udrea 2014; Katz et al. 2018) to our suggested way of evaluating on the original task. Following the previous work, we set the switching criteria to switch once an increase of 20% in expanded nodes was observed. Table 1 presents a pairwise comparison of $K^*$ domain-level performance in terms of coverage for $k = 1000$ with the four heuristics, with and without transformation back to original task. The latter is denoted by adding ↓ to heuristic name. The last row of the table shows the overall coverage for each of the configurations. Every other entry in the table represents the number of domains where the row configuration achieves better coverage than the column one. First, we observe that evaluating the heuristic on the original task significantly improves the performance, across all tested heuristics. The most significant improvement is observed for CEGAR, where there are 28 domains with improved coverage, compared to 2 domains with reduced coverage, and overall in-

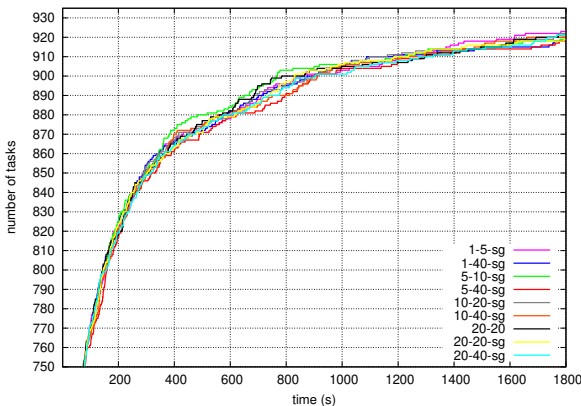

Figure 3: Any-time performance of *selected* LMcut configurations. Label lb-ub-sg: lower bound, lb, upper bound, ub, and considering reaching goal state for switching, sg.

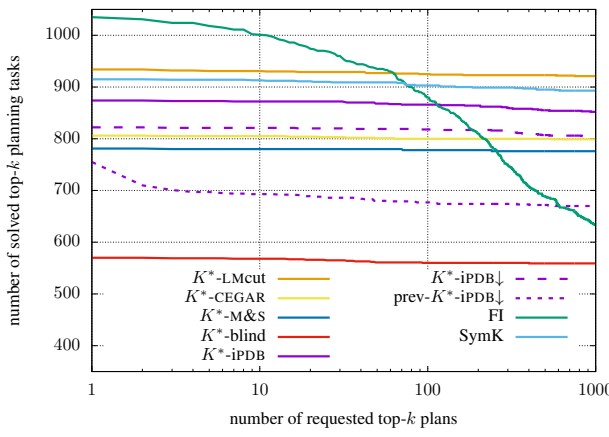

Figure 4: Any-k performance showing the number of top-$k$ planning tasks solved per varying $k$.

crease of 78 tasks solved. Second, note that the inconsistent heuristic configuration LMcut is the best performer overall, showing the benefit of an algorithm that supports the use of inconsistent heuristics.

For the next experiment, we focus on LMcut, evaluated on the original task. For $k = 1000$, we test the influence of the decisions on switching between the $A^*$ and $EA$, in an any-time manner. The switching criteria involve lower and upper bounds, as well as whether a goal state was reached again. For the bounds in $\{1, 5, 10, 20, 30, 40, 50, 100\}$, assuming the lower bound being strictly smaller than the upper bound, together with the historical configuration of single bound of 20 (labeled 20-20), as in the previous experiment. This results in the total of 58 configurations. Figure 3 shows the any-time coverage of selected configurations. Note, while their performance is somewhat similar, for smaller timeouts of 300–1000 seconds the difference in coverage can be significant, with the configuration 20-20 rarely being the top performer.

Finally, going back to the four heuristics from the first experiment, we show the any-k behavior of these configurations, comparing to other top-k planning approaches, bidirectional symbolic blind search (SymK) (Speck, Mattmüller, and Nebel 2020) [1] and Forbid Iterative planner (FI) (Katz et al. 2018), as well as previous implementation of a $K^*$ variant (Katz et al. 2018), depicted in Table 1 and Figure 4 by the prev-K*-iPDB↓ label. For the iPDB heuristic (purple lines, solid and dashed), we can observe a significant improvement in the performance when switching to our variant of $K^*$. Some of it can be attributed to evaluating the original task, but even with iPDB↓ (dashed lines) we see an improvement to the any-k behavior. It is worth noting that SymK is no longer the top performer. The overall coverage of SymK is 892, compared to 921 of $K^*$ with LMcut. Per-domain comparison reveals that $K^*$ with iPDB ties with SymK 30 to 30, and $K^*$ with LMcut achieves superior performance on 24 domains vs. 28 domains in favor of SymK.

---

[1]As in other approaches, the $h^2$ preprocessor was switched off.

Going back to our new variant of $K^*$, note that for all heuristics, the any-k performance is more level now, with the difference between top-1 and top-k being 13 for LMcut, 22 for iPDB, 7 for CEGAR, 5 for M&S, and 11 for blind heuristic. As top-1 is essentially cost-optimal planning with $A^*$, this is the upper bound on $K^*$ performance achievable without improving $A^*$.

## Conclusions and Future Work

In this paper, we revisit $K^*$ search algorithm for top-$k$ planning, and renovate $K^*$ search in various aspects, from simplified yet optimized implementation of data structures to algorithm configuration study that reflects the behavior of $A^*$ search. We also observed a significant impact of task transformation that might degrade the quality of heuristics and show how to mitigate such a negative impact.

The empirical evaluation results encourage further development of $K^*$ search approaches in planning. In earlier evaluations, $K^*$ was the worst performer compared with other approaches. However, the presented methods were able to elevate the performance of $K^*$ to almost reach its theoretical ability, achieving state-of-the-art performance. The overall performance of $K^*$ is bounded by the performance of $A^*$, where numerous techniques are ready to be explored in the context of $K^*$ search. For future work, we would like to take inspiration from methods improving $A^*$ performance through search space pruning, such as partial order reduction (Wehrle and Helmert 2012) and symmetry reduction (Domshlak, Katz, and Shleyfman 2012). Another possible interesting avenue of research is how to improve the memory efficiency of $K^*$.

$K^*$ and $EA$ are generic algorithm that can be applied to wider range of problems. Looking at broader scopes, $K^*$ search could find applications in other planning tasks, such as top-quality planning, or even in a non-optimal setting, such as diverse planning. In addition, simplified yet efficient implementation of $EA$ and $K^*$ may be applied to other problem domains, such as combinatorial optimization or approximation algorithms for the counting tasks.

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
