# OpenReview forum: "On K* Search for Top-k Planning"
_icaps-conference.org/ICAPS/2023/Workshop/HSDIP — ICAPS HSDIP 2023_

### Official Review · Reviewer_zCw8 · 2023-04-13
**Clear, correct and significant, with minor room for improvement**

**Rating:** 8
**Confidence:** 3

**Review:**

**Summary**
This paper studies the K* search algorithm in the context of Top-k planning. Several changes to the algorithm are proposed and evaluated empirically, including a revision of internal data structures, alternative sound criteria under which the search alternates between A* search and Eppstein's algorithm, and an improved method to evaluate heuristics in presence of a potentially necessary single-goal task transformation.

The paper is well-structured and presents its contributions in a mostly clear and understandable way. However, there is room for some improvements regarding the presentation (see **Notes on Clarity** below). To the best of my knowledge, the contributions are formally correct and novel. The experimental evaluation highlights the most important aspects and shows that the made adjustments to the algorithm significantly improve over previous implementations of the algorithm and make K* search competitive with state of the art planners. However, some things are left unclear. It is unfortunately unclear to me which algorithm configurations for K* were used in the last comparison. I assume the traditional switching criteria from A* to EA (20-20) with improved heuristic evaluation and data structures is used here. The configuration of the previous K* implementation is however still unclear to me, and with it the value of the comparison. A clear description of this configuration is missing and should be included. From what I assume, this configuration may to use the previous heap data structures instead of linked lists and build the path graph for EA explicitly, but this is not specified further.

I have three questions/suggestions for the authors:
- Traversing linked lists is usually a very slow operation, but must eventually be done by Eppstein's algorithm. Is there a reason to use linked lists in the implementation other than for keeping the references to $\mathsf{STE}$ entries valid? In that case, C++'s std::deque seems like a more appropriate data structure to me, keeping references stable and also allowing for traversal close to a normal array.
- Do you have any data regarding the quality of the considered heuristics when evaluated on the task transformation vs the original task? For example, the accuracy of the initial state estimate.
- Did you measure the impact of the new data structures in isolation, with the other algorithm parameters fixed? (perhaps K*-iPDB vs prev-K*-iPDB represents this comparison?)

**Notes on Clarity**
- I think the notation $\mathsf{STE}(u, v)$ is poorly introduced, which lead to me mistaking $\mathsf{STE}$ as a function. The property of being a sidetracked edge should first be defined as $(u, v) \in G_{A*} \setminus T_{A*}$, and afterwards $\mathsf{STE}(u, v)$ should be introduced as a convenience notation to mean an edge $(u, v)$ with this property.
- The algorithm parameters rs-EA and sog were hard to parse and confused me upon first reading. They should be written in a dedicated font. Also, dashes in a parameter are not ideal, and it did not become clear to me what sog (switch on goal) stands for until Algorithm 5.
- In the definition of $\delta(u, v)$: Can't there be multiple operators with different costs that lead from $u$ to $v$?
- In the paragraph about list-based $H_{in}$ and $H_T$, one could mention that **sorted** linked lists are used instead
- I did not understand the sentence "Note that K* can utilize $EA_{Thr}$ when it switches from A* to EA".

**Miscellaneous**
- **Input:** and **Output:** could be used instead of **Require:** and **Ensure:** in algorithms.
- One could add to Figure 1 (caption): ... over the path from $S_0$ to $v$ (**in** $\mathbf{T_{A*}}$)

**Typos**
- There a missing indentation for **then** in Algorithm 3
- Abstract: of K* based approach -> of *the* K* based approach (re-appears twice in the paragraph before the background section)
- Abstract: on classical benchmark -> on classical benchmark*s*
- Introduction (column 2): results clearly motivate*s* -> results clearly motivate
- Background (column 2): familiarity of A* -> familiarity *with* A*
- Algorithm 1: Init*ai*lize -> Init*ia*lize
- Figure 1 (caption): Illustration of *of* -> Illustration of, in linked list -> *as (a)* linked list(s)
- Page 5 (column 1): implmented -> impl*e*mented
- Page 5 (column 2): ... valid top-k plans when its child ... Should this mean its head children?
- Page 5 (column 2): sin*lg*e -> sin*gl*e
- Page 5 (column 2): missing subscripts (s1 and s4)
- Page 6 (column 2): heuristic*s* function -> heuristic function

---

> ### Author Response · Authors · 2023-05-03
> **Thanks for your review.**
>
> Thanks very much for your reviews.
> We will do our best to update the draft with your notes on clarification.
>
> For the first question on the std::list, deque does not guarantee the validity of the iterator.
> We used iterator objects while implementing the successor generator of EA,
> and std::list met that requirement.
> It is more than keeping the reference to the side-tracked-edge objects stable.
> std::list does not sort items automatically like heaps, but we utilize some aspects in A* to avoid sorting lists every time we change the list.
>
> The meaning of the sentence, "Note that K* can utilize $EA_{Thr}$...",
> is that when we know the previous $EA_{Thr}$, which is the head value of A* queue that prevented EA from extracting more solution further, A* can decide to switch to EA to extract at least one more solution by checking its head value of queue with that previous $EA_{Thr}$.

---

> > ### Author Response · Authors · 2023-05-04
> > **Experiment configurations and previous K* implementation**
> >
> > In the experiment, other than comparing the variations in the hyper-parameters, all results are reported based on the 20-20 configuration.
> > The previous K* implementation used a vector-based approach. As mentioned before, due to the changes in Hin and Htree in K*, there may be better choices for handling such changes than vector-like containers. It was only working with a consistent heuristic. In earlier days, people also considered that K* might only work with consistent heuristics. It isn't easy to precisely compare the two implementations as many factors influence the performance. Still, roughly, 20-20 configuration would be comparable between new-K*+iPDB and old-K*-iPDB to see the improvement against the past one. But, replacing the vectors in the previous K* with lists will differ from the new k* implementation.

---

> > > ### Author Response · Authors · 2023-05-04
> > > **Data regarding the quality of the heuristics**
> > >
> > > While we do have the data regarding the initial state heuristic values, we did not make the comparison.
> > > We will consider including such a comparison in the final version if space permits and interesting insights are visible.

---

### Official Review · Reviewer_EBhL · 2023-04-25
**Well Suited for the Workshop**

**Rating:** 7
**Confidence:** 4

**Review:**

Summary:

In this paper, K* is revisited, which is a heuristic search algorithm for finding the k best, i.e., the k shortest or k least-cost solutions in a given state space. The underlying motivation for revising K* is that, despite well-known complexity results in graph theory for K* (when graphs are explicitly given), the performance of K* for planning problems has been rather weak compared to other approaches. The paper analyzes why this is the case and shows, among other things, that the data structures used in standard K* implementations of planning software can be a bottleneck and are not the originally described ones with which the K* complexity results hold. Therefore, this paper describes how to close this gap in the implementation of planning software and identifies further customization and design options. An empirical evaluation shows that this modified version of K* and the new implementation, which is closer to the way K* was originally presented, is much stronger than its predecessor. Furthermore, it is competitive with the state of the art for top-k planning and advantageous in several domains.

Overall assessment:

The topic of the paper is well suited for the workshop, and the results presented are interesting. The paper has some weaknesses in presentation, as it is rather technical in the main part, where intuition for how the underlying algorithm and data structures used work is sometimes lacking (see detailed feedback). However, in my opinion, this kind of work is well suited for the workshop, as it highlights and revisits a rather important heuristic algorithm. Overall, I recommend that the paper be accepted.


Detailed Feedback:

1. Presentation:
Overall, the paper is well structured, but the more technical parts are hard to follow in places. Starting with the paragraph "Eppstein's Implicit Path Representation". As described in the paper, it presents the algorithms of Eppstein (1998) and Aljazzar and Leue (2011) in a potentially more accessible and high-level way. However, the description is not really high-level, and in places it is quite technical. Of course, it is less technical than the original works, but I think these technical sections would benefit greatly from additional, even higher-level descriptions.

2. Data Structures:
Partly because of the technical descriptions, I could not quite figure out whether the implementation presented in the paper and used for the experiments now uses the heap structures presented by Eppstein (1998) and Aljazzar and Leue (2011), or the new linked-list representation proposed in the paper. I think this should be made clearer, and if linked lists are used, it raises the question of whether the nice complexity results of the original papers still hold. Maybe this implementation is still not as good as it could be? Although it is better than the original implementation used by the planning community to date.

3. Code Release:
Since one motivation of the paper is to present a more sophisticated and well-performing implementation of K* for planning problems, it would be an important and valuable part to release the implementation so that future research can reproduce and compare the results. Do you plan to release the code?

4. Minors:
- For SymK it is never mentioned which search strategy is used. Probably it is the default and overall strongest one: "symbolic bidirectional blind search". I would mention this in the paper.
- Fast Downward is highlighted in small caps, but no other planner is (inconsistent).

---

> ### Author Response · Authors · 2023-05-03
> **Thanks for your reviews.**
>
> Thanks very much for your reviews.
> We will do our best to accommodate your feedback in the paper to improve the clarity.
> We will take the process of open-sourcing the code and hope to finish the process soon to release the code.
>
> In the paper, we implemented the new linked-list-based data structures.
> The original complexity result of EA does not hold when we do the worst-case analysis due to the fact that increased complexity for sorting the linked list. As explained in the paper, the successor generator of EA traverses internal nodes in those heaps.
> Unlike EA, the underlying heap data structures for EA keep changing in K* as we expand nodes in A*.
> Therefore, properly handling pointers is as important as using heap data structures.
> This issue all boils down to whether the pointers (iterators in C++) directing the nodes in Hin and Htree  (they are not heaps anymore, and H may mislead readers, but we decided to maintain the same notation in K* paper) remain valid after we modify the data structure.
> Vector-like data structure does not guarantee the validity of iterators.
> When we tested vector-based implementation earlier, the overhead of fixing the change in K* was too much that it lost all the benefit of using EA (the flat curve as shown in Figure 4, essentially the time difference of finding one plan and 1000 plans is within 30 min time out).

---

### Decision · Program_Chairs · 2023-05-05

**Decision:**

Accept

**Comment:**

We are happy to announce that the paper has been accepted for the workshop.

Both reviewers clearly expressed their support. Please make sure to incorporate the feedback mentioned by the reviewers in the final version.